# Probiotics-Containing Yogurt Ingestion and *H. pylori* Eradication Can Restore Fecal *Faecalibacterium prausnitzii* Dysbiosis in *H. pylori*-Infected Children

**DOI:** 10.3390/biomedicines8060146

**Published:** 2020-06-01

**Authors:** Yao-Jong Yang, Peng-Chieh Chen, Fu-Ping Lai, Pei-Jane Tsai, Bor-Shyang Sheu

**Affiliations:** 1Departments of Pediatrics, National Cheng Kung University Hospital, College of Medicine, National Cheng Kung University, Tainan 70428, Taiwan; yaojong@mail.ncku.edu.tw (Y.-J.Y.); marvelous3918@gmail.com (F.-P.L.); 2Institute of Clinical Medicine, National Cheng Kung University Hospital, College of Medicine, National Cheng Kung University, Tainan 70428, Taiwan; pengchic@mail.ncku.edu.tw; 3Departments of Pathology, National Cheng Kung University Hospital, College of Medicine, National Cheng Kung University, Tainan 70428, Taiwan; peijtsai@mail.ncku.edu.tw; 4Department of Medical Laboratory Science and Biotechnology, College of Medicine, National Cheng Kung University, Tainan 70428, Taiwan; 5Departments of Internal Medicine, National Cheng Kung University Hospital, College of Medicine, National Cheng Kung University, Tainan 70428, Taiwan; 6Internal Medicine & Institute of Clinical Medicine, Kaohsiung Medical University, Kaohsiung 80708, Taiwan

**Keywords:** *H. pylori*, *F. prausnitzii*, gut microbiota, probiotics, next generation sequencing

## Abstract

This study investigated the compositional differences in fecal microbiota between children with and without *H. pylori* infection and tested whether probiotics-containing yogurt and bacterial eradication improve *H. pylori*-related dysbiosis. Ten *H. pylori*-infected children and 10 controls ingested probiotics-containing yogurt for 4 weeks. Ten-day triple therapy plus yogurt was given to the infected children on the 4th week. Fecal samples were collected at enrollment, after yogurt ingestion, and 4 weeks after successful *H. pylori* eradication for cytokines and microbiota analysis using ELISA and metagenomic sequencing of the V4 region of the *16S rRNA* gene, respectively. The results showed *H. pylori*-infected children had significantly higher levels of fecal TGF-β1 than those who were not infected. Eight of 295 significantly altered OTUs in the *H. pylori*-infected children were identified. Among them, the abundance of *F. prausnitzii* was significantly lower in the *H. pylori*-infected children, and then increased after yogurt ingestion and successful bacterial eradication. We further confirmed probiotics promoted *F. prausnitzii* growth in vitro and in ex vivo using real-time PCR. Moreover, *F. prausnitzii* supernatant significantly ameliorated lipopolysaccharide-induced IL-8 in HT-29 cells. In conclusions, Probiotics-containing yogurt ingestion and *H. pylori* eradication can restore the decrease of fecal *F. prausnitzii* in *H. pylori*-infected children.

## 1. Introduction

*Helicobacter pylori* (*H. pylori*) infection primarily starts in childhood [1]. This microorganism has been reported to cause peptic ulcers, gastric lymphoma, and adenocarcinoma mainly in adults, and also iron deficiency anemia and growth retardation in children [2,3,4,5]. A case-control study of a Japanese cohort showed that acquisition of *H. pylori* in early life increased the risk of developing both gastric cancer and gastric ulcers [6]. Most consequences may originate from local chronic gastric inflammation [7,8,9]. However, knowledge regarding the relationship between *H. pylori* infection and gut inflammatory parameters is still lacking.

Patients with inflammatory bowel diseases have been shown to have loss of gut microbiota diversity, which referred to as dysbiosis [10,11]. A meta-analysis review article concluded that the bacterial count of *Faecalibacterium prausnitzii* in patients with inflammatory bowel diseases was significantly lower than that in healthy controls [12]. This negative impact of an aberrant microbiota on health may be attributed to chronic inflammation [13].

The association between gastric microbiota diversity and *H. pylori* infection is still under debate [14,15,16]. In human and rhesus macaque stomach, the presence of *H. pylori* did not significantly affect the composition of the gastric community [14,15]. In contrast, Aebischer et al. reported that *H. pylori* infection causes an increase in microbial diversity in a mouse model [16]. Our previous study showed that *H. pylori*-infected children had a lower ratio of the amount of fecal *Bifidobacterium* spp. and *Bifidobacterium* to *E. coli* than non-infected controls [17]. This condition has also been concomitantly found with a reduction in IgA level in *H. pylori*-infected children. As secretory IgA (sIgA) plays an essential role to regulate gut microbiota [18], a chronic *H. pylori* infection may drive dysbiosis of the gut microbiota and contribute to the occurrence of local and systemic disorders. Accordingly, a more powerful tool is still needed to define the changes at a microbiome genus level.

Probiotics and probiotics-containing yogurt are popular health-promoting foods with beneficial effects on the gastrointestinal inflammations [17,18,19,20,21]. Our previous study showed that probiotics-containing yogurt ingestion could reduce gastric *H. pylori* load, increase systemic IgA level and fecal *Bifidobacterium* spp./*E. coli* ratio in children [18]. Moreover, we further demonstrated that *Lactobacillus acidophilus* could ameliorate *H. pylori*-induced inflammation in gastric epithelium by inactivating the Smad7 and NFκB pathways [20]. Gamallat et al. have reported that *L. rhamnosus* could ameliorate intestinal NFκB and TNFα and prevent colon cancer development in an animal model [21]. We thus test whether probiotics can be as effective as *H. pylori* eradication to restore the changes of the gut microbiota in the *H. pylori*-infected children.

## 2. Results

### 2.1. Prevalence of H. pylori Infection

A total of 179 children from primary schools were enrolled. The mean age was 10.7 years with a male to female ratio of 1.1. The seroprevalence of *H. pylori* infection was 12.8%, and there was no significant difference between the male and female children (*p* = 0.36). In addition, there were no significant differences in body weight (45.0 vs. 41.4 kg, *p* = 0.19) and height (148.0 vs. 145.1 cm, *p* = 0.22) between the children with and without *H. pylori* infection.

### 2.2. Numbers of Study Cases and Fecal Samples

The numbers of enrolled subjects and fecal samples are shown in Figure 1. A total of 37 stool samples including *H. pylori*-infected (HPS1, *n* = 10), *H. pylori*-infected with yogurt (HPS2, *n* = 6), *H. pylori*-infected with yogurt and eradication (HPS3, *n* = 7), non-*H. pylori* infected controls (CS1, *n* = 9), and controls with yogurt (CS3, *n* = 5) were processed for DNA purification, 16S-rRNA gene amplification, and amplicon sequencing. A stool sample from a control subject (CS1) was excluded from the analysis as the child had consumed yogurt twice a week prior to sample collection. Another fecal sample from HPS2 was not analyzed owing to a low yield of DNA.

### 2.3. Fecal Inflammatory Parameters between H. pylori-Infected Children and Controls

The fecal samples were tested for calprotectin, lactoferrin, IL-6, TGF-β1, and sIgA (Appendix A). The results showed that the *H. pylori*-infected children had a significantly higher fecal TGF-β1 level (12.0 vs. 7.0 ng/mL, *p* = 0.02) than the non-infected controls. In addition, the fecal calprotectin (13.2 vs. 2.5 μg/g, *p* = 0.13) and lactoferrin (15.4 vs. 6.8 μg/g, *p* = 0.18) levels were also higher in the *H. pylori*-infected children than in the non-infected children, although with only marginal significance. The sIgA level was lower in the *H. pylori*-infected children than in the controls (240.3 vs. 505.0 μg/mL, *p* = 0.07), however there was no significant difference in IL-6 level (3.4 vs. 1.3 pg/mL, *p* = 0.27) between the two groups.

### 2.4. Sequencing Results

16S rRNA sequencing reads were classified to 295 OTUs from the 37 fecal samples. In phylogenetic analysis, the fecal bacterial community mostly belonged to five major phyla, including Bacteroidetes (54%), Firmicutes (32.8%), Proteobacteria (7%), Actinobacteria (3.3%), and Fusobacteria (0.1%). *H. pylori* could only be identified in 60% of the samples from the infected patients by 16S rRNA sequencing. The dendrogram of hierarchical clustering and proportional changes in bacterial OTU abundance at the genus level is shown in Figure 2A. The most abundant genus in the 37 samples was *Bacteroides* spp., followed by *Prevotella* spp. and *Faecalibacterium* spp. However, there were no significant differences in the microbiota composition between the five groups in AMOVA (*p* = 0.331) and HOMOVA (*p* = 0.522) analysis.

### 2.5. Library Coverage and Sequence Diversity

The rarefaction curve, a comparison of α diversity between subjects categorized by HPS1, HPS2, HPS3, CS1, and CS3 showed high biodiversity between the *H. pylori*-infected and control groups (Figure 2B). The slope of each curve reached a plateau with increasing sequence depth. The inverse Simpson index was significantly different between CS1 and CS3, but not between HPS1, HPS2, and HPS3 (Figure 2C). Weighted UniFrac phylogenetic distance matrices were used to calculate the β diversity, which was shown in PCoA plots. In comparisons of group diversity between CS1 and HPS1, CS1 and CS3, and HPS1, HPS2, and HPS3, and all groups both AMOVA and HOMOVA showed no significant difference (*p* > 0.05) in microbial diversity between groups (Appendix A).

### 2.6. Significant Genus Difference between H. pylori-Infected Children and Controls

In comparisons of significant microbial changes in the phylum (Appendix A) and genus (Table 1) levels between the *H. pylori*-infected (HPS1) and control (CS1) subjects, the mean abundance ratio of *Proteobacteria* was significantly higher in HPS1 than in CS1 at the phylum level (0.044 vs. 0.026, *p* = 0.02).

In addition, we identified eight OTUs with significant changes in relative abundance at the genus level between the two groups by *Metastats*. The *H. pylori*-infected children had a significantly lower mean proportional abundance of the fecal microbes *Faecalibacterium prausnitzii* (OUT004, the only species), *Porphyromonadaceae* (OUT065), *Desulfovibrio* spp. (OUT051), and *Eubacterium* spp. (OUT122), and significantly increased *Alphaproteobacteria* (OUT082), *Coriobacteriaceae* (OUT070), *Howardella* spp. (OUT136), and *Pseudoflavonifractor* spp. (OUT063). Only *F. prausnitzii* had a mean proportional abundance > 0.02.

### 2.7. Serial Changes in the Abundance of F. prausnitzii after Yogurt Ingestion and H. pylori Eradication

Figure 3A shows the serial changes in the abundance of *F. prausnitzii* in the five groups. In addition to a significant reduction in the *H. pylori*-infected children (HPS1) compared to the controls (CS1), probiotics-containing yogurt ingestion significantly increased the proportion of *F. prausnitzii* in the non-*H. pylori* infected children (0.11 vs. 0.05, *p* = 0.003). However, this increase was diminished in the *H. pylori*-infected group after yogurt ingestion (0.04 vs. 0.02, *p* = 0.11). Moreover, the mean proportional abundance of *F. prausnitzii* in the *H. pylori*-infected children with successful eradication was significantly increased compared to that before yogurt ingestion and eradication (0.10 vs. 0.02, *p* = 0.0005). The abundance level of *F. prausnitzii* after *H. pylori* eradication was comparable to that of the non-infected level. Real-time PCR quantification confirmed the increase in the abundance of fecal *F. prausnitzii* in the samples (Figure 3B).

### 2.8. Probiotics Facilitate F. prausnitzii Growth in Vitro and In Ex Vivo

To identify the probiotics directly or indirectly facilitate *F. prausnitzii* growth, we tested the *F. prausnitzii* growth condition with relative DNA level by real-time PCR in vitro and in ex vivo. Figure 4A showed that culture supernatants of *L. acidophilus* and *B. lactis* directly promoted *F. prausnitzii* growth in a co-culture system. The enhanced effect was disappeared, if the cecum fluid collecting from C57B/6 mice was selected instead of Brain Heart Infusion Supplement (BHIS) media (Figure 4B). Furthermore, the cecum fluids of mice had positive effect on promotion of *F. prausnitzii* growth when compared to BHIS culture media (Figure 4C). Moreover, the supernatant of *L. acidophilus* or/and *B. lactis* cultured with cecum tissue (ex vivo) of mice induced a higher relative ratio of *F. prausnitzii* growth than culture fluid of extracorporeal cecum tissue (Figure 4D).

### 2.9. F. prausnitzii Ameliorated LPS-Induced IL-8 Expression in HT-29 Cells

To evaluate the functional benefits of probiotics in restoring the proportion of fecal *F. prausnitzii* associated with *H. pylori* infection, AGS and HT-29 cells were treated with supernatants of *F. prausnitzii* culture for 4 h before *H. pylori* or LPS inoculation. Figure 5 shows that *F. prausnitzii* supernatant treatment significantly abolished LPS-induced IL-8 production in the HT-29 cells at 6 h (Figure 5A). However, *F. prausnitzii* supernatant did not affect the *H. pylori*-induced IL-8 expression in the AGS cells (Figure 5B).

## 3. Discussion

This study demonstrated a significant reduction of fecal *F. prausnitzii* in *H. pylori*-infected children. Moreover, both probiotics-containing yogurt ingestion and *H. pylori* eradication with yogurt ingestion improved the abundance of fecal *F. prausnitzii*. Based on in vitro and ex vivo assessment, supplement of *L. acidophilus* and *B. lactis* can facilitate *F. prausnitzii* growth, which can offer benefit to decrease the LPS-induced gut inflammation.

Gut sIgA is a first-line barrier to play an important role in the regulation of host-microbiota homeostasis [22,23]. Our previous study showed *H. pylori* infection in children significantly reduced serum IgA levels [17]. In this study, we again found that the *H. pylori*-infected children had an obvious reduction in sIgA than the non-infected controls. The TGF-β and TGF-β receptor signaling are important for systemic and mucosal IgA production [24,25,26]. Our data was compatible to show a significant increase in fecal TGF-β of the *H. pylori*-infected children due to negative feedback to sIgA depletion. Concerning the fecal calprotectin concentration in *H. pylori*-infected children, our results being compatible with others showed gastric *H. pylori* infection did not increase fecal inflammatory parameters [9,27]. The insignificant correlation may cause by wild variation of inflammatory markers between individuals. Further study need to more strict control confounding factors, which may affect microbiota and gut inflammation in study subjects.

Previous reports have shown that *H. pylori* colonization of the stomach alters the diversity and richness of other gastric microbiomes [28,29]. This study is the first to investigate differences in fecal microbiota between *H. pylori*-infected and non-infected children using next generation sequencing. We identified eight various genus/species which were significantly different in abundance between the two groups. Among them, only *F. prausnitzii* was relatively abundant in the gut. This bacterium is one of the most abundant commensal bacteria, and it has been shown to produce a large amount of butyrate in the guts of humans and other animals [30,31]. Clinically, the depletion of fecal *F. prausnitzii* may serve as a biological marker in patients with inflammatory bowel diseases. This is the first study to report a close association between pediatric *H. pylori* infection and depletion of gut *F. prausnitzii*. These results highlight the importance of *H. pylori* and *F. prausnitzii* in gut inflammatory diseases. Further studies are needed to clarify how *H. pylori* colonization in the stomach influences colonic microorganisms.

The ingestion of probiotics or probiotics-containing yogurt is beneficial to human health. The positive effects may be through modulation of the gut microbiota, immune function, and metabolomics [17,32]. Although this study has disclosed that fecal inflammatory parameters (calprotectin, lactoferrin, and IL-6) increased in *H. pylori*-infected children than controls, it is lack for the follow-up data in children after eradication and yogurt ingestion. Few studies have reported that probiotics-containing yogurt ingestion can increase the levels of certain commensal organisms, which are beneficial to gut health. Because the study yogurt containing *Lactobacillus* and *Bifidobacterium*, we are interested in whether the yogurt ingestion can increase the abundance of fecal *Lactobacillus* and *Bifidobacterium* in children. As shown in the Appendix A, the relative ratio of *Bifidobacterium* increased in non-*H. pylori* infected children with yogurt ingestion. However, supplement of yogurt did not increase the fecal *Bifidobacterium* colonization in *H. pylori*-infected children. Unfortunately, the abundance of *Lactobacillus* was very rare even children having yogurt ingestion. Our study has shown yogurt ingestion really promotes gut *F. prausnitzii* growth in children with and without *H. pylori* infection. Moreover, our in vitro and ex vivo study also confirmed that *L. acidophilus* and *B. lactis* can enhance *F. prausnitzii* growth via direct (co-culture) and indirect (probiotics-intestine culture supernatant) manners. Finally, in agreement with other studies, we confirmed that *F. prausnitzii* supernatant can ameliorate LPS-induced inflammatory cytokines in HT-29 cells [33,34,35]. Further studies are needed to elucidate the relationship between *H. pylori* infection and reduction in gut *F. prausnitzii* abundance and to investigate the functional proteomics and metabolomics of *F. prausnitzii* and the effects on gut inflammatory diseases.

The major strength of the study is that the in vivo experiments utilizing the clinical samples identify the microbiome prominently in *H. pylori*-infected children. Moreover, to demonstrate the probiotics facilitate *F. prausnitzii* growth by clinical trial, ex vivo, and in vitro experiments. However, there are several limitations in this study, including the small number of participants limits the strength of this finding. Second, how *H. pylori* colonization in the stomach influences gut sIgA and cytokines that alters the abundance of colonic microorganisms is not investigated. Third, we do not study the other significantly changed microorganisms with tiny amount after *H. pylori* infection and yogurt ingestion. Whether these bacteria play positive or negative roles desire futher investigations. Finally, what components or species of probiotics can regulate gut homeostasis is unclear.

In conclusions, *H. pylori*-infected children have a dysbiosis in gut *F. prausnitzii,* which can be restored by *H. pylori* eradication and probiotics-containing yogurt. Probiotics-containing yogurt can also increase the abundance of *F. prausnitzii* in the non-infected children. The effect of the restored abundance of *F. prausnitzii* is anticipating to the control of gut inflammation.

## 4. Materials and Methods

### 4.1. Subject Inclusion and Exclusion Criteria

This study enrolled students from four elementary schools in Tainan City, Taiwan. Their age ranged from 10 to 12 years. After obtaining consent (A-BR-102-105 approval from the Institutional Review Board, National Cheng Kung University Hospital on 18 February 2014) from each individual and their parents, information on underlying diseases, *H. pylori* infection status, antibiotics intake, yogurt (probiotics) consumption, and H_2_-blocker or proton pump inhibitor use was recorded. Children who had known major organic diseases such as immunodeficiency disorders, malignancy, and diseases treated with chemotherapy, steroids and antibiotics (within 1 month), predisposed to the influence of gastrointestinal microbial colonization and immunological function were excluded. The children who ingested probiotics or probiotics-containing yogurt with a frequency of more than twice per week 1 month prior to enrollment were dropped out from fecal examinations.

### 4.2. Serum Collection and Diagnosis of H. pylori Infection

Overnight fasting blood samples (6–8 mL) were drawn from each participant after obtaining consent from the children and their parents. The serum was then tested for anti-*H. pylori* IgG antibodies (HEL-p TEST^TM^ II; AMRAD Biotech, Perth, Western Australia) using an enzyme-linked immunosorbent assay (ELISA) with a sensitivity and specificity > 90% [36]. The seropositive and control children were further confirmed by ^13^C-UBT to diagnose ongoing *H. pylori* infection. The cutoff value of positive ^13^C-UBT was defined as an excess ^13^CO2 ⁄ ^12^CO2 ratio of more than 4.0‰ [37].

### 4.3. Probiotics-Containing Yogurt Ingestion, H. pylori Eradication and Follow-Up

The ^13^C-UBT-confirmed *H. pylori*-infected children and age- and sex-matched controls ingested one bottle of AB-yogurt twice daily for 4 weeks. The yogurts (200 mL per bottle) containing at least 5 × 10^9^ live organisms (*Lactobacillus acidophilus*, *Bifidobacterium lactis*, *Lactobacillus bulgaricus*, and *Streptococcus thermophilus*). The yogurt ingestion procedure followed our previous study [17]. To improve the compliance of yogurt consumption, school nurses directly watched and recorded its drinking at 07:30 a.m. and 03:30 p.m. on school days. On the weekend, the parents were requested to record those details on the same paper. After the course was completed, returned records were analyzed for compliance. Good compliance was defined as at least 80% consumption of total designed amount. Thereafter, the *H. pylori*-infected children received triple therapy for 10 days (pantoprazole: 2 mg/kg/day, max. 40 mg bid, amoxicillin: 50 mg/kg per day, max. 1 g bid, and clarithromycin: 15 mg/kg per day, max. 500 mg bid) with yogurt. The controls took yogurt only for 10 days. The success or failure of eradication therapy was tested by ^13^C-UBT 4 weeks after completing treatment and yogurt ingestion in the *H. pylori*-infected children.

### 4.4. Stool Collection and Preparation

Two thumb-sized fresh stool samples each weighing approximately 0.2–0.4 g (HPS1 and CS1) were collected from each participant in the morning. A second stool sample (HPS2) was collected from the *H. pylori*-infected group 4 weeks after yogurt ingestion. Third fecal samples (HPS3 and CS3) were collected at the completion of the 4 weeks of treatment and/or yogurt ingestion in both groups. Total fecal DNA was extracted using a Qiagen stool kit (Qiagen, Chatworth, CA, USA) and then stored as −20 °C. The other sample was diluted in PBS containing 2.5 mg/mL leupeptin, 11 mg/mL aprotinin, and 0.5 mM 4-(2-aminoethyl) benzenesulfonyl fluoride (Sigma, St. Louis, MO, USA). After thorough mixing and centrifugation for 10 min at 10,000 *g*, the supernatant was stored as −80 °C for further ELISA tests.

### 4.5. Analysis of Fecal sIgA, IL-6, TGF-β, Lactoferrin, and Calprotectin

The prepared supernatant was tested by ELISA for IL-6, IL-10, and TGF-β (Quantikine; R&D Systems, Minneapolis, MN, USA). Fecal lactoferrin and calprotectin were measured using a commercial Leuko-Test kit (TechLab, Blacksburg, VA, USA) and PhiCal Fecal Calprotectin Immunoassay kit (Genova Diagnostics, Asheville, NC, USA), respectively, following the manufacturers’ instructions.

### 4.6. Amplicon Sequencing (16S-rRNA) for Fecal Microbiota Diversity

Multiplex bar-coded indexes were used for paired-end sequencing of the *16S rRNA* variable region 4 (V4). We used the universal bacterial primer pairs 515F (5′-TCGTCGGCAGCGTCAGATGT GTATAAGAGACAGGTGCCAGCMGCCGCGGTAA-3′) and 806R (5′-GTCTCGTGGGCTCGGAGA TGTGTATAAGAGACAGGGACTACHVGGGTWTC TAAT-3′) for maximal coverage of bacterial phylogeny [38]. Additional appropriate barcode and linker oligonucleotides were appended to the primer pairs. Illumina amplicon library generation was performed as described previously, except for the additional steps of purification of the PCR products with AMPure (Beckman Coulter, Brea, CA, USA) and quantification using a Qubit fluorometer (Invitrogen Life Technologies, London, UK) and quantitative PCR (qPCR) (Kapa Biosystems, Wilmington, MA, USA). The amplified bar-coded DNA from 119 samples was then pooled. The samples were diluted to a final dilution of 10 pM, combined at a 90:10 ratio with 10 pM of balancing library, and run with a 2 × 250 cycle reaction on an illumina MiSeq platform (Illumina, San Diego, CA. USA). Sequences of *16S rRNA* were aligned and analyzed with standard operation protocol by *mothur* software v.1.39 (https://www.mothur.org/ wiki/MiSeq_SOP).

### 4.7. Probiotics Facilitate F. prausnitzii Growth In Vitro and Ex Vivo

*F. prausnitzii* (APC 918/95b) was purchased from the Institute of Food Science and Technology, National Taiwan University. Bacteria were cultured using YCFA GSC medium and broth in absolute anaerobic conditions [39]. *Lactobacillus acidophilus* and *Bifidobacterium lactis* (LA5^®^ and Bb12, originated from the Chr. Hansen, Denmark, provided by the President Corp., Tainan, Taiwan) was used. They were maintained on a Brucella agar, incubated in anaerobic conditions.The cecum fluid and tissue were obtained form 8-week-old C57B/6 mice. The cecum tissue was cut flat using sterile scissors, and the equal-sized tissue was co-cultured with 3 mL RPMI medium with or without *L. acidophilus* and *B. lactis* in dishes for 24 h. The supernatant was filtered through a 0.22 μL filter, and then added to quantitative *F. prausnitzii* and YCFA for co-culture. After the 4th, 8th, and 12th hour of co-culture, the relative abundance of *F. prausnitzii* were measured as the percentage of total bacteria DNA copy numbers by real-time PCR using the specific primers [40].

Gastric AGS and colonic HT-29 cells (3 × 10^6^ cells/well) were prepared and pretreated with supernatants for 4 h. Clinically isolated *H. pylori* (HP238, MOI = 100) and LPS (100 ng/mL) were then added to the dishes for serial time periods. The supernatants were collected at various time points and centrifuged for IL-8 analysis by ELISA (R&D, Minneapolis, MN, USA).

### 4.8. Statistical Analysis

All analyses were performed with built-in commands in *mothur*. Alpha diversity was determined by rarefaction curves describing the number of operational taxonomic unit (OTUs) with inverse Simpson diversity estimates. Differences in alpha diversity were assessed using the t-test and a repeated measure ANOVA. Beta diversity was assessed using unweighted and weighted UniFrac distance matrices and visualized using principal coordinate analysis (PCoA) [41]. Statistically significant differences in the relative abundance of OTUs between variable groups of patients were analyzed using Metastats (https://www.mothur.org/wiki/metastats/).

## Figures and Tables

**Figure 1 biomedicines-08-00146-f001:**
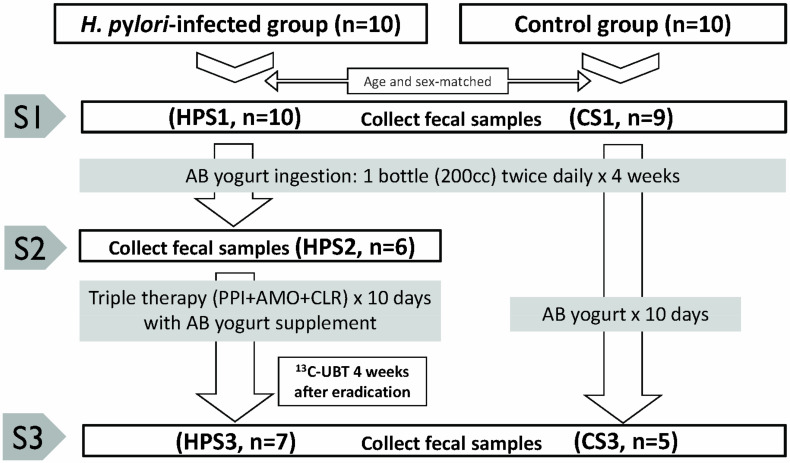
The numbers of subjects and fecal samples at enrollment, on the 4th week, and 4 weeks after triple eradication therapy. CS1: non-*H. pylori* infected controls at enrollment; CS3: non-*H. pylori* infected controls 4 weeks after yogurt ingestion; HPS1: *H. pylori*-infected children at enrollment; HPS2: *H. pylori*-infected children after 4 weeks of yogurt ingestion; HPS3: *H. pylori*-infected children 4 weeks after eradication therapy.

**Figure 2 biomedicines-08-00146-f002:**
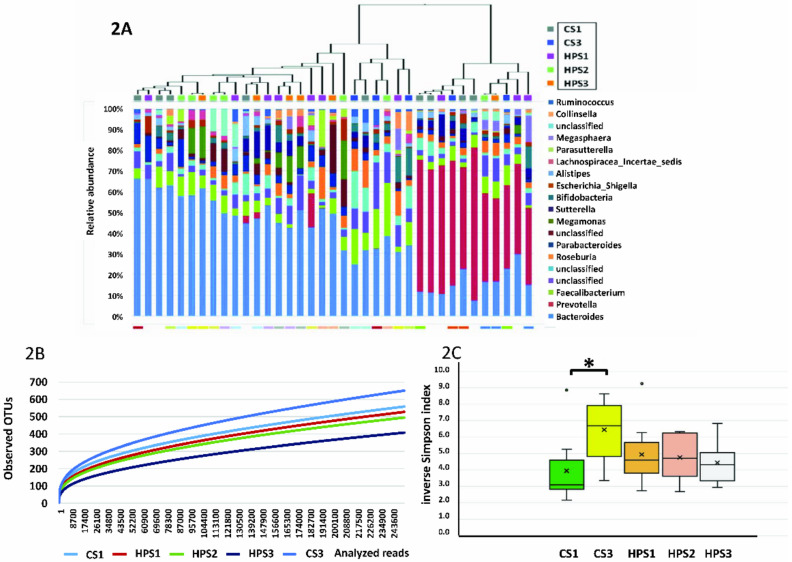
The dendrogram of hierarchical clustering and microbiota composition at the genus level in the 37 fecal samples (**A**). The rarefaction curves (**B**) and inverse Simpson index (**C**) of microbiota between the five groups (CS1: non-*H. pylori* infected controls at enrollment; CS3: non-*H. pylori* infected controls 4 weeks after yogurt ingestion; HPS1: *H. pylori*-infected children at enrollment; HPS2: *H. pylori*-infected children after 4 weeks of yogurt ingestion; HPS3: *H. pylori*-infected children 4 weeks after eradication therapy. * *p* < 0.05.

**Figure 3 biomedicines-08-00146-f003:**
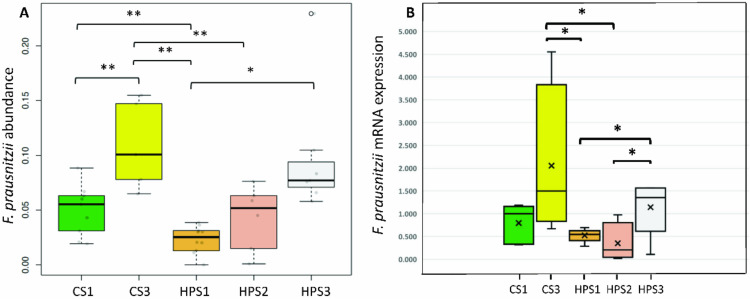
Serial changes in the abundance of *F. prausnitzii* (OTU004) using Metastats (**A**) and real-time RT-PCR (**B**) analysis. (* *p* < 0.05, ** *p* < 0.001).

**Figure 4 biomedicines-08-00146-f004:**
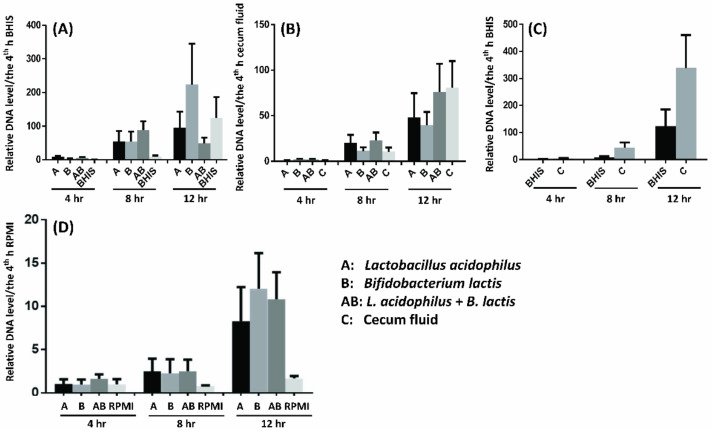
*Lactobacillus acidophilus* and *Bifidobacterium lactis* promote *F. prausnitzii* growth in vitro and in ex vivo. All relative DNA level were calculated by tested value/the 4th h control value. The *L. acidophilus* and *B. lactis* were sub-cultured in BHIS broth and incubated until the bacterial density reached OD ≥ 0.5 (5 × 10^7^/μL). Then 100 μL of culture media was added in 5 mL RPMI-1640 (**A**) or 20 μL in 1 mL cecum fluid (**B**) and incubated for 24 h. After filtration, 100 μL of supernatant was added in 30 mL YCFA broth with *F. prausnitzii* (APC 918/95b) co-cultured for 4, 8, and 12 h. (**C**) *F. prausnitzii* cultured in YCFA with BHIS or cecum fluid only. (**D**) The equal-sized cecum tissue was co-cultured with 3 mL RPMI medium with or without *L. acidophilus* and *B. lactis* for 24 h. The 100 μL supernatants was added in 30 mL YCFA broth with *F. prausnitzii* co-cultured for 4, 8, and 12 h. The relative abundance of *F. prausnitzii* were measured as the percentage of total bacteria DNA copy numbers by real-time PCR. All tests were triplet.

**Figure 5 biomedicines-08-00146-f005:**
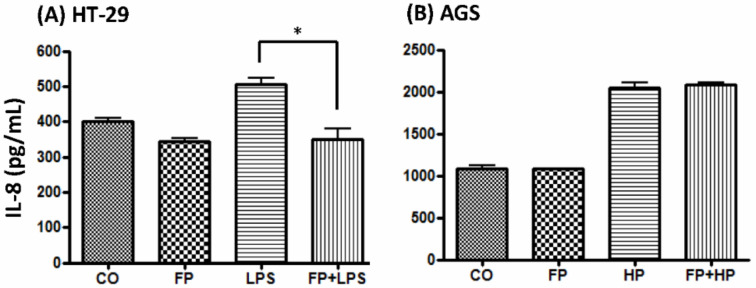
The effects of pretreatment with *F. prausnitzii* supernatant on IL-8 levels in LPS-treated HT-29 cells (**A**) and *H. pylori*-treated AGS (**B**). The IL-8 levels were tested at 6 h. CO: control cells; FP: treatment with *F. prausnitzii* supernatant; HP: treatment with *H. pylori*; LPS: treatment with LPS (10 μg/mL); FP + LPS: pretreatment with *F. prausnitzii* supernatant and then LPS was added; FP + HP: pretreatment with *F. prausnitzii* supernatant and then *H. pylori* was added. * *p* < 0.05.

**Table 1 biomedicines-08-00146-t001:** The mean proportional abundance of eight genes with significant differences in the feces between *H. pylori*-infected children and controls.

OTUs; Taxonomy	Mean Proportional Abundance (%)		Percent Change	*p* Value
	*H. pylori*-Infected (HPS1, *n* = 10)	Controls (CS1, *n* = 9)		
004; Firmicutes; Clostridia; Clostridiales; Ruminococcaceae; Faecalibacterium	2.3347	5.0	−2.14	0.005994
065; Bacteroidetes; Bacteroidia; Bacteroidales; Porphyromonadaceae	0.0004	0.1859	−464.75	0.008991
051; Proteobacteria; Deltaproteobacteria; Desulfovibrionales; Desulfovibrionaceae; Desulfovibrio	0.002	0.2433	−121.65	0.015984
122; Firmicutes; Clostridia; Clostridiales; Eubacteriaceae; Eubacterium	0	0.0031	−	0.005351
082; Proteobacteria; Alphaproteobacteria	0.1157	0	−	0.000999
070; Actinobacteria; Actinobacteria; Coriobacteriales; Coriobacteriaceae	0.0101	0	−	0.003996
136; Firmicutes; Clostridia; Clostridiales; Lachnospiraceae; Howardella	0.0036	0	−	0.004299
063; Firmicutes; Clostridia; Clostridiales; Ruminococcaceae; Pseudoflavonifractor	0.0653	0.0013	0.02	0.007992

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
