# Peer review of "Probiotics-Containing Yogurt Ingestion and H. pylori Eradication Can Restore Fecal Faecalibacterium prausnitzii Dysbiosis in H. pylori-Infected Children"

_biomedicines, 2020, doi:10.3390/biomedicines8060146_

Round 1

Reviewer 1 Report

The authors Yang et al. investigated the compositional changes in fecal microbiome in children with and without H. pylori infection. They used probioitic containing yogurt to assess the changes. To do this, they utilized 37 subjects and divided into 5 groups including controls and collected fecal samples and performed 16S rRNA sequencing and also looked at fecal inflammatory markers such as calprotecting, lactoferrin, TGF-b1 and sIgA. They found significant reduction in F. praustnitzii levels in H.pylori-infected children.

The major strength of the study is use of clinical as well as invivo experiments utilizing the clinical samples. As observed in current literature, yet again 16S rRNA seq is proved to identify the microbiome prominently, not forget the increasingly important role of probiotics for health. 

Major limitations of the study are: although they have good number of samples, still it may need further research. Also, this gives us limited chances on extrapolating these findings. Perhaps authors can include a section of strengths and limitations.

Before going for publication, please follow my comments below:

What are the inclusion and exclusion criteria with regards to any prior dairy products (milk, cheese or fermented products) and fibre-intake which are potential confounders.

Perhaps, looking at any microbial translocation could have been beneficial, but not essential.

Authors should be precise in mentioning the number of CFU/ml of each probiotic in the yogurt??

Did authors take note of disease activity index?

Which approach did authors use to analyze 16S rRNA data? which tool have they used?

Figure 2A quality can be improved or potentially change to percent abundances?

In the Supp, fig S1, authors showed Beta diversity profiles, however failed to compare all the groups together? Perhaps that addition will be beneficial to compare along with the alpha diversity profiles in fig 2c.

Did authors see any visible changes in Lactobacillus and Bifido counts post probiotic treatment, report the changes, as whether it was only F. prausnitzii or other genera also altered.  Authors can utilize supplementary section to showcase all the observed genera.

Reviewer 2 Report

Thank you very much for let me review this manuscript by Yao-Jong Yang et al.

The study investigated whether probiotics-containing yogurt ingestion and Helicobacter pylori eradication can restore fecal Faecalibaterium prausnitzii dysbiosis in H.pylori-infected children.

The topic is very interesting. The study is substantially well-conducted. However, the small number of participants limits the value of this work. Moreover, to make more attractive and clearer the reading of the manuscript, the structure and overall style of the manuscript need to be improved.

The manuscript needs an extensive English revision throughout the text. 

Abstract

L34-35: methods and results are mixed up, confusing the period.

Introduction

L53-54: please develop this period (2-3 sentences) with the corresponding references.

L61-62: please develop this period (2-3 sentences) with the corresponding references.

L66-67: This is a conclusion statement, not to be put in the introduction.

Materials and Methods

Please insert the Methods part.

The research manuscript sections of this journal are the following: Introduction, Materials and Methods, Results, and Discussion.

L244-250: Please specify the composition of the AB-yogurt. Ther is no information regarding the probiotic composition of the yogurt. Moreover, how and when (in the morning? Afternoon?) did children ingest the yogurt?

Results

Figure 1: the sample size is short at the baseline and it furtherly falls down at S3. It is difficult to infer something in these cases.

L70-75: A table could be more appropriate to clearly describe the characteristic data of the control and intervention groups.

Figure 1: The abbreviations should be specified.

Table 1 is interesting but needs to be re-worked to make it more readable. It appears to report raw data.

Discussion

The discussion is confusing and too superficial and should be re-worked to make it more substantial and attractive.  The results of the study need to be challenged.

L199 and L206: please eliminate “to the best of our knowledge, this study is  ..”

L202: please delete “using Metastats”

L211-212: Please develop the positive effects of probiotics-containing yogurt

In a paragraph, please indicate the strengths and limitations/bias of the study.

Round 2

Reviewer 2 Report

The manuscript has improved significantly